# Microwave photon Fock state generation by stimulated Raman adiabatic passage

Shavindra P. Premaratne[1,2], F.C. Wellstood[1,3] & B.S. Palmer[1,2]

The deterministic generation of non-classical states of light, including squeezed states, Fock states and Bell states, plays an important role in quantum information processing and exploration of the physics of quantum entanglement. Preparation of these non-classical states in resonators is non-trivial due to their inherent harmonicity. Here we use stimulated Raman adiabatic passage to generate microwave photon Fock states in a superconducting circuit quantum electrodynamics system comprised of a fixed-frequency transmon qubit in a three-dimensional microwave cavity at 20 mK. A two-photon process is employed to overcome a first order forbidden transition and the first, second and third Fock states are demonstrated. We also demonstrate how this all-microwave technique can be used to generate an arbitrary superposition of Fock states. Simulations of the system are in excellent agreement with the data and fidelities of 89%, 68% and 43% are inferred for the first three Fock states respectively.

[1] Department of Physics, University of Maryland, College Park, Maryland 20742, USA. [2] Laboratory for Physical Sciences, College Park, Maryland 20740, USA. [3] Joint Quantum Institute and Center for Nanophysics and Advanced Materials, College Park, Maryland 20742, USA. Correspondence and requests for materials should be addressed to S.P.P. (email: shavi@umd.edu).

Quantum states with a well-defined number of quanta, or Fock states, have many roles in quantum information processing including quantum key distribution[1], quantum memory[2], generation of identical photons for remote entanglement[3] and universal control of quantum elements[4]. There have been a number of proposals for generating Fock states[5,6] within the framework of cavity quantum electrodynamics, and the first experimental demonstration involved exchanging an excitation from a Rydberg atom to a coupled cavity[7].

Fock states were first generated in a circuit quantum electrodynamics (QED) system by coupling a tunable superconducting qubit to a microwave resonator[8,9]. In these experiments, the excited state of the superconducting qubit was tuned to be resonant with the resonator and the excitation from the qubit was then transferred to the resonator by vacuum Rabi oscillations. An advantage of this technique is that the transfer can be performed in the relatively fast time of 25 ns; a necessary criterion since tunable qubits are prone to sources of dephasing such as magnetic flux noise. More recently, arbitrary Fock states were generated using selective number dependent arbitrary phase gates on a fixed-frequency qubit-cavity system[4,10]. This approach took advantage of a highly coherent qubit-cavity system to generate the desired states in a time of 1 μs.

In this article, we report the use of a two-photon Stimulated Raman Adiabatic Passage (STIRAP) protocol to transfer the excitation from a fixed-frequency transmon qubit to a microwave cavity Fock state in 354 ns. STIRAP is a technique to transfer the population from an initial state $|0\rangle$ to a final state $|2\rangle$ via an intermediate state $|1\rangle$ by applying two Gaussian coherent Raman pulses that overlap in time: an initial one on the $|1\rangle$ to $|2\rangle$ transition and the second one on the $|0\rangle$ to $|1\rangle$ transition[11,12]. STIRAP overcomes a forbidden transition (for example, $|0\rangle$ to $|2\rangle$ to first order), while never occupying the intermediate state and is robust against slight variations in the drive parameters[11]. Previous proposals and demonstrations of STIRAP using superconducting quantum devices involved transfer of populations between the three lowest levels of the device[13–15]. The qutrit levels have a relatively high anharmonicity and this simplifies the implementation. In contrast, here we employ STIRAP to operate on composite cavity–qubit states, a system that has a small amount of anharmonicity. Using a STIRAP protocol with the third excited state of the qubit–cavity system as the intermediate state, we transfer the first excited state population of the qubit to the cavity and thereby create a single-photon Fock state. By performing driven coherent manipulations on the generated state as well as multi-level simulations, the fidelity for the Fock state is determined to be $\geq 85\%$, a value similar to other initial demonstrations[8–10]. This process is extended to create the second and third Fock states, with simulated fidelities of 68% and 43%, respectively. These fidelities are limited by the decay rate of the cavity relative to the adiabatic process duration. Finally, we demonstrate how this protocol can be used to create a superposition of Fock states.

## Results

**Circuit QED system.** Our system consisted of a fixed-frequency transmon qubit embedded in a three-dimensional superconducting aluminium cavity[16] that is cooled to 20 mK on a Leiden CF-450 dilution refrigerator (Fig. 1a). The qubit[17] had a transition frequency $\omega_q/2\pi = 5.5297$ GHz between the ground and excited state. It was fabricated on a sapphire substrate and had a single Al/AlO$_x$/Al Josephson junction shunted by two large Al pads that reduced its charging energy (Fig. 1b). The large pads also provided

coupling with strength $g/2\pi = 69.8$ MHz to the TE$_{101}$ fundamental mode of the cavity with a resonance frequency $\omega_c/2\pi = 7.9370$ GHz (refs 16,17). Since the coupling strength is much smaller than the qubit–cavity detuning ($g/(\omega_c - \omega_q) \ll 1$), the undriven qubit–cavity system is well described by the dispersive Jaynes–Cummings Hamiltonian[18]:

$$\mathcal{H}_{JC} = \frac{1}{2}\hbar\omega_q\sigma_z + \hbar(\omega_c + \chi\sigma_z)\hat{\mathcal{N}} \qquad (1)$$

where $\sigma_z$ is the Pauli matrix operating on the state of the qubit, $\hat{\mathcal{N}}$ is the cavity photon number operator and $\chi/2\pi = -2.0$ MHz is the frequency dispersive shift resulting from the cavity–qubit coupling.

The system's energy level structure from equation (1) has two ladders, with one ladder for the ground state of the qubit $|g\rangle$ and the second ladder for the excited state of the qubit $|e\rangle$ (Fig. 1c). The rungs of each ladder correspond to a different number of cavity photons. To drive transitions between different states in this energy level structure, microwaves were delivered to the input port of the cavity with a coupling $Q_{in} \sim 400,000$. The output port with coupling $Q_{out} \sim 200,000$ sent the microwaves to a low noise high-electron-mobility transistor (HEMT) amplifier at 3 K to measure the cavity resonance and the state of the qubit. The state of the system was interrogated using a high-power cavity transmission measurement that was sensitive to the transmon state (Supplementary Fig. 1)[19]. Averaging repeated measurements yielded the total probability ($\mathcal{P}_e$) to find the system in the $|e\rangle$ ladder.

The transmon had a characteristic energy lifetime of $T_1 = 24$ μs and a Ramsey decay time of $T_2^* = 22$ μs while the cavity had a lifetime of $\kappa^{-1} = 2.5$ μs set by $Q_{out}$ for purposes of measurement. Note that our device operated in the strong dispersive regime of circuit QED since $2\chi \gg \left\{\kappa, T_1^{-1}, \left(T_2^*\right)^{-1}\right\}$ (ref. 20).

**Fock state generation.** The generation of the first Fock state $|g1\rangle$ involved the four lowest levels in the system: $|g0\rangle$, $|e0\rangle$, $|g1\rangle$ and $|e1\rangle$ (Fig. 1c). After passively cooling the system to the $|g0\rangle$ state, a 32 ns duration $\pi_{g0\to e0}$ pulse at the qubit frequency $\omega_q$ was sent to add one excitation to the qubit and obtain the state $|e0\rangle$. To transfer population from $|e0\rangle$ to $|g1\rangle$, which is a first-order forbidden transition, STIRAP was used via the intermediate level $|e1\rangle$ (Fig. 1d). In this protocol, a Stokes tone was first applied at frequency $\omega_{s1} = \omega_q + 2\chi - \delta_1$ slightly detuned by an amount $\delta_1$ from the $|g1\rangle$ to $|e1\rangle$ transition frequency and with a Gaussian envelope for the amplitude. We next applied a slightly delayed Gaussian-shaped pump tone pulse that overlapped with the Stokes pulse. The pump pulse had a frequency $\omega_{p1} = \omega_c + 2\chi - \delta_1$, which included the same detuning $\delta_1$ as the Stokes tone.

For a conventional three-level system undergoing STIRAP, process fidelity does not depend heavily on the detuning from the intermediate level. However, in our multilevel system, a negative detuning from level $|e1\rangle$ was beneficial to minimize leakage to higher levels and was optimized through simulations. The counterintuitive profile for these two pulses was

$$\Omega(t) = \Omega_s \exp\left[-\frac{(t-t_s)^2}{2\sigma^2}\right]\sin[\omega_{s1}t + \phi_s] + \Omega_p \exp\left[-\frac{(t-t_p)^2}{2\sigma^2}\right]\sin[\omega_{p1}t + \phi_p]$$

$$(2)$$

where $\Omega_s$ and $\Omega_p$ were the peak amplitudes of the Stokes and pump tones, respectively, $\sigma$ was the width of the Gaussian envelope for both pulses, $t_s$ and $t_p$ were the times when the Stokes and pump envelopes are at their maximum, and $\phi_s$ and $\phi_p$ were offset phases in the Stokes and pump carrier tones, respectively (Supplementary Table 3).

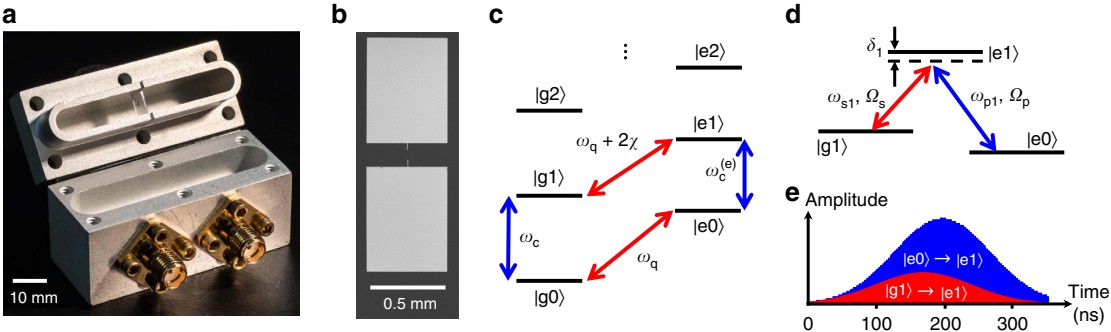

**Figure 1 | STIRAP in a transmon–cavity system.** (**a**) Photograph of transmon chip in the top of opened 3D Al cavity. (**b**) Optical micrograph of the Al/AlO$_x$/Al transmon on sapphire chip. (**c**) Level structure (not to scale) for the transmon–cavity system. Letters and numbers within kets denote qubit and cavity photon-like states, respectively. (**d**) Levels used for the generation of $|g1\rangle$ Fock state with the Stokes (red) and pump (blue) tones with frequencies $\omega_{s1}$, $\omega_{p1}$ and drive strengths $\Omega_s$, $\Omega_p$, respectively. $\delta_1$ is the common detuning for both drives from $|e1\rangle$. (**e**) Pulse envelopes for the Stokes (red) and pump (blue) tones.

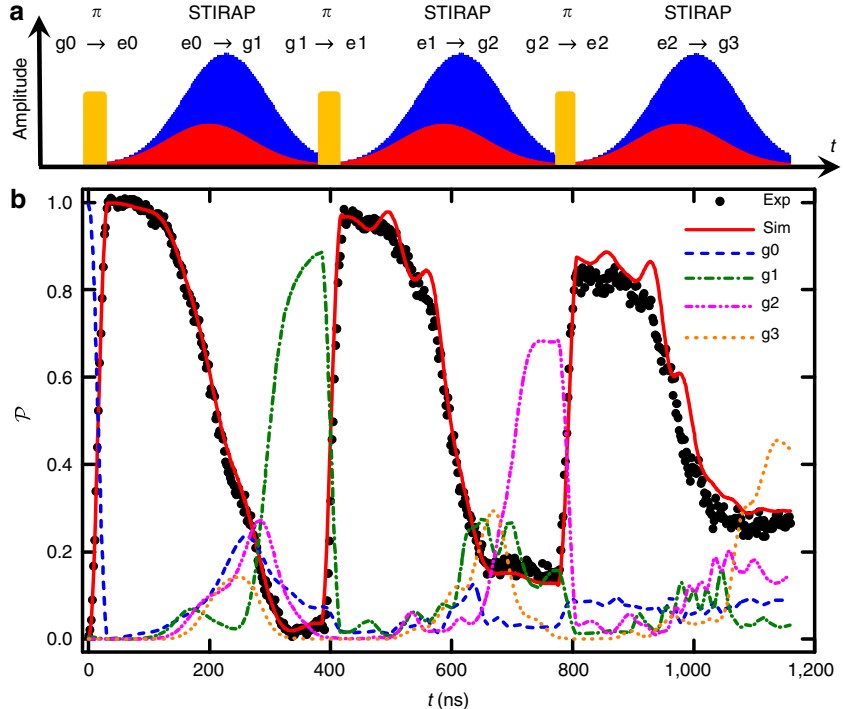

**Figure 2 | Generation of the first three Fock states.** (**a**) Pulse sequence for generating Fock states $|g1\rangle$, $|g2\rangle$ and $|g3\rangle$. The $\pi$-pulses (yellow) transfer $|g0\rangle \rightarrow |e0\rangle$, $|g1\rangle \rightarrow |e1\rangle$ and $|g2\rangle \rightarrow |e2\rangle$. The Stokes (red) and pump (blue) pulses transfer $|e0\rangle \rightarrow |g1\rangle$, $|e1\rangle \rightarrow |g2\rangle$ and $|e2\rangle \rightarrow |g3\rangle$. (**b**) State probabilities ($\mathcal{P}$) versus time (t) for $|g1\rangle$, $|g2\rangle$ and $|g3\rangle$ Fock state generation protocol shown in **a**. Measured data are total qubit excited state populations $\mathcal{P}_e$ (black dots). Solid red curve shows the corresponding $\mathcal{P}_e$ from density matrix simulations. Dashed/dotted curves show simulated evolution of the populations of Fock states $|g0\rangle$, $|g1\rangle$, $|g2\rangle$ and $|g3\rangle$ corresponding to 0, 1, 2 and 3 photons in the cavity, respectively.

To determine the initial choices for STIRAP parameters, density matrix simulations were performed at the outset of the experiment. We solved the master equation in the laboratory frame while applying the time dependent STIRAP protocol. Following initial experiments, the model was refined using a total of 21 levels in the two ladder system. The programmed waveforms used for the experiment were used in performing the numerical simulations as well (see Methods).

Figure 2 shows the measured probabilities $\mathcal{P}_e$ (filled points) of the qubit being in its excited state versus time during the execution of the STIRAP protocol generating the first, second and third Fock states. For comparison, results from the simulations are shown as curves. For the data in this plot, execution of the protocol was halted every 1 ns at which point $\mathcal{P}_e$ was destructively measured. After measuring $\mathcal{P}_e$, the protocol was started again from the beginning. For the first 32 ns, a $\pi_{g0 \rightarrow e0}$ is sent to the device and upon completion $\mathcal{P}_e > 96\%$. After placing the system in $|e0\rangle$, the Stokes and pump STIRAP pulses were sent to the device using $\sigma = 70.8$ ns, $t_p - t_s = 14$ ns, a Raman detuning of $\delta_1 = 8.1$ MHz from the $|e1\rangle$ state, $\Omega_s/2\pi = 9.6$ MHz, and $\Omega_p/2\pi = 26.2$ MHz. The drive phases had no effect on the generation of this state and for simplicity were set to zero $\phi_{p1} = \phi_{s1} = 0$. The total duration to complete STIRAP was 354 ns and from simulations, the fidelity upon completion was $\mathcal{F}_{g1} = 89\%$, limited by photon leakage out of the cavity.

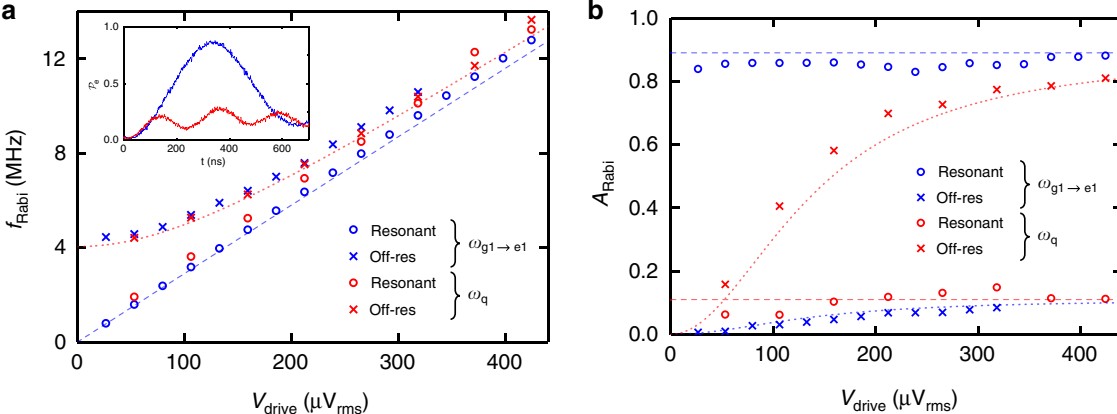

**Figure 3 | Verification of |g1⟩ Fock state.** (**a**) Extracted Rabi frequencies versus drive amplitude following generation of |g1⟩. Dashed and dotted lines depict expected frequencies for a two-level system subjected to an on-resonant and a 4.04 MHz detuned off-resonant drive, respectively. Inset shows driven Rabi oscillations at $\omega_{g1 \to e1}$ (blue) and $\omega_q$ (red) frequencies, at a relatively small drive amplitude of $V_{drive} = 53\,\mu V_{rms}$. (**b**) Extracted Rabi amplitude oscillations. Dashed and dotted lines depict expected behaviour for a two-level system after setting up the cavity in a Fock state with $\mathcal{F}_{g1} = 89\%$.

Due to the relatively small photon anharmonicity and the relatively short lifetime of photons in the cavity, Wigner tomography[21] could not be performed in our system. Instead, the fidelity of the Fock state was demonstrated by performing coherent Rabi oscillations immediately following the STIRAP protocol. The Rabi oscillations were performed at both the |g1⟩ → |e1⟩ transition frequency ($\omega_{g1 \to e1}$) and at the |g0⟩ → |e0⟩ transition frequency ($\omega_q$) for a range of drive amplitudes (Supplementary Figs 6–8). For small drive amplitudes $V_{drive}$, the oscillations when driven at $\omega_q$ (Fig. 3a inset) show a larger Rabi frequency $f_{Rabi}$ and lower $\mathcal{P}_e$ amplitude, consistent with the drive frequency being detuned. Each Rabi oscillation was fit to the sum of two oscillatory functions to extract the frequency ($f_{Rabi}$) and the amplitude $A_{Rabi}$ of each component. The amplitude of the resonant component is a metric for the initial state fidelity.

Figure 3a shows the four extracted Rabi frequencies when the system was driven at the two drive frequencies $\omega_{g1 \to e1}$ and $\omega_q$. Two of the data sets are linear with drive voltage and extrapolate to a zero Rabi frequency at zero drive voltages, consistent with a resonant drive process. The other pair of Rabi frequencies extrapolates, at small drive voltages, to a frequency of ∼4 MHz, consistent with a non-resonant process with detuning $2\chi$ (Fig. 1c). The fidelity of the |g1⟩ Fock state can be determined from the amplitude of the Rabi oscillations (Fig. 3b). At small drive voltages, the amplitude for the resonant component when driving at $\omega_{g1 \to e1}$ shows a fidelity of $\mathcal{F}_{g1} \geq 85\%$. In comparison, when simulating a two-photon red sideband transition[22] with these device parameters, the best fidelity obtained was only $\mathcal{F}_{g1} = 65\%$, a value significantly lower than the result from the STIRAP protocol (Supplementary Fig. 11).

Following generation of |g1⟩, the second (|g2⟩) and third (|g3⟩) Fock states were produced in a similar manner (Fig. 2). For each step, the parameters were tuned to optimize the fidelity of the desired state (Supplementary Table 3). |g2⟩ was created by adding another excitation to the system: a $\pi_{g1 \to e1}$ pulse was followed by another STIRAP protocol taking the system to |g2⟩ with a simulated fidelity $\mathcal{F}_{g2} = 68\%$ (Supplementary Fig. 9). A similar process was implemented to generate |g3⟩ with a simulated fidelity $\mathcal{F}_{g3} = 43\%$. As with |g1⟩, simulations revealed that $\mathcal{F}_{g2}$ and $\mathcal{F}_{g3}$ were limited by $\kappa$.

For comparison, the red curve in Fig. 2 shows the simulated results for the excited state probability. Except for some small, fine-scale oscillatory behaviour found in the simulation during the |g2⟩ and |g3⟩ STIRAP protocol, there is good

agreement with the experiment. The dashed and dotted curves show the expected populations of the various Fock states in the cavity. The rotating wave approximation was not employed in the simulations and the 21-level simulation for |g1⟩ generation was also checked against a simulation performed with 51 levels in the two-ladder system.

To verify that this is a STIRAP process, we note that Bergmann et al.[11] derive both local and global criteria for adiabatic following in a system. The local adiabaticity criterion addresses the instantaneous non-adiabatic coupling possible with various pulse shapes. As this experiment employed smooth Gaussian pulses, the local adiabaticity criterion was naturally satisfied. The global adiabaticity criterion addresses adiabaticity for the entire process. This condition for STIRAP is given by

$$\Omega_{eff}\Delta\tau > 10 \qquad (3)$$

where $\Omega_{eff} = \sqrt{\Omega_P^2 + \Omega_s^2}$ is the root-mean-square (r.m.s.) Rabi frequency and $\Delta\tau$ is the total overlap time for the two pulses[11]. Using equation (2) and our parameters for generating |g1⟩, we found $\int \Omega_{eff}\,d\tau \sim 30$, satisfying global adiabaticity. The relative insensitivity of the STIRAP protocol to parameter fluctuations was verified through simulations (Supplementary Table 4).

**Superposition of Fock states.** STIRAP can also be used to create more exotic states such as superpositions of Fock states. To generate a superposition of |g0⟩ and |g1⟩, the initial $\pi_{g0 \to e0}$ pulse for the Fock state creation was replaced with a 16 ns $(\pi/2)_{g0 \to e0}$ pulse to create the state $(|g0\rangle + |e0\rangle)/\sqrt{2}$. Next the |e0⟩ → |g1⟩ STIRAP protocol coherently transferred the population in |e0⟩ to |g1⟩ (Fig. 4a). Following the 370 ns total execution time, we observed $\mathcal{P}_e \approx 0$ and simulations showed that the state resides in an equal superposition of |g0⟩ and |g1⟩.

During the STIRAP protocol, we found that $\mathcal{P}_e$ exceeds 0.5 and displays an oscillatory pattern (Fig. 4a). To understand this aspect of the data, the phase of each microwave drive was independently varied. The oscillations depended strongly on the phase $\phi_{\pi/2}$ of the initial $\pi/2$ pulse and the phase $\phi_s$ of the Stokes tone but not the phase $\phi_P$ of the pump tone (Supplementary Fig. 4). This phase dependence and the fact that $\mathcal{P}_e$ exceeds 0.5 during execution suggests that the population in |g0⟩ from the initial state plays a pivotal role when turning on the Stokes tone. Once again, excellent agreement is found between the data and the simulations. The simulations suggest that the final state has been prepared with a fidelity of ∼93%, limited by the decay rate $\kappa$ of the cavity.

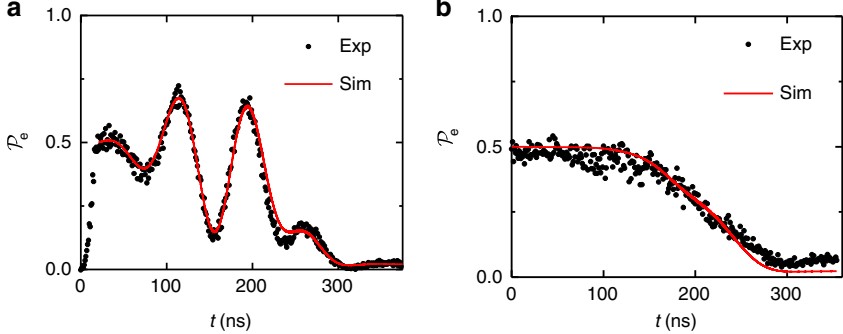

**Figure 4 | Generation of superpositions of Fock states with STIRAP.** (**a**) Equal superposition between |g0⟩ and |g1⟩ generated using STIRAP after a $\pi/2$ initialization pulse. All phases for the drives and the initialization were set to zero ($\phi_{\pi/2} = \phi_{\rm s} = \phi_{\rm p} = 0$). (**b**) STIRAP performed on the system after driving to a mixed steady state with a 150 μs long pulse at frequency $\omega_{\rm q}$. Experimental data points (black circles) are overlaid with the simulated pooled populations (solid red).

To further understand the effect of the initial state, STIRAP was performed after the qubit was driven to a mixed steady state, removing qubit phase coherence. This resulted in the disappearance of oscillations observed previously (Fig. 4b). We infer that the oscillations were the direct result of coherent interference within the system when undergoing STIRAP. We believe the four lowest levels |g0⟩, |e0⟩, |g1⟩ and |e1⟩ set up a suitable system for observing quantum interference between difference excitation pathways[23], though confirming the source of this interference is beyond the scope of this work. The existence of oscillations throughout the interaction demonstrates the coherent nature of the interactions as well as the preservation of coherence during execution.

## Discussion

A protocol based on STIRAP was demonstrated for coherently transferring the population from a superconducting qubit to $n = 1$, 2 and 3 Fock states in a superconducting microwave cavity. The number of drives to implement this protocol is $3n$ from the ground state, where $n$ is the target Fock state.

On the basis of our simulations, increasing the lifetime of the cavity by a factor of 10 (ref. 24) would improve the fidelity for generation of the first Fock state from the present value of 89% to 95%. An increased lifetime would also allow the application of more sophisticated analysis techniques on the generated states[10] that were not viable in the present system due to low anharmonicity. Further improvements in the Fock state fidelity would require increasing the width of the STIRAP Gaussian pulses ($\sigma$). Alternatively, increasing the dispersive shift $\chi$ would enable the use of stronger drives and shorter STIRAP times as the increased anharmonicity reduces leakage to other states. Implementing adiabatic shortcuts[25–27] is another approach that could potentially speed up the transfer process in our system. We also note that if the cavity lifetime is much shorter than the qubit lifetime, STIRAP can be used to reset the qubit by transferring population from the |e⟩ ladder to the |g⟩ ladder and using the short cavity lifetime to initialize to |g0⟩.

This approach also works in the reverse direction, so that if the order of the Stokes and probe pulses are reversed then the population will be transferred from the cavity to the qubit. We have used this technique to measure the lifetime of the first Fock state and find it to be 2.6 μs, which is in good agreement with the decay rate of the cavity (Supplementary Fig. 10). Such a protocol could be used for single-shot detection of itinerant microwave photons with noise limited only by the fidelity of the qubit readout. Such a high-fidelity single-photon detector may allow measurements of the Bell state between the qubit and the cavity (Supplementary Fig. 5) including violation of Bell's inequality[28].

## Methods

**Device fabrication.** The cavity (dimensions: 5 mm × 21.6 mm × 35 mm) was machined from 6063 aluminium and then etched for 3 h at 50 °C using commercial Al etchant. The input and output coupling quality factors ($Q_{\rm in}$ and $Q_{\rm out}$) were set and measured at room temperature by measuring the scattering parameters $S_{11}$, $S_{21}$ and $S_{22}$.

The transmon was fabricated on a 5 mm × 7 mm sapphire substrate by electron-beam lithography using a double resist stack of ZEP-520 A and MMA(8.5)MAA EL11. Following development, Al was thermally evaporated at two angles ± 12.5° (from the chip surface's normal) for thicknesses of 30 and 50 nm, respectively. Prior to the second evaporation, oxidation was performed for 210 s at an O₂ pressure of 118 mTorr. Following the second evaporation, the device was passivated in pure oxygen for 20 min at 260 mTorr. Finally, lift-off of the unexposed resist was performed in hot Microposit-1165.

**Experimental set-up.** The cavity–transmon system was attached to the mixing chamber of a Leiden Cryogenics CF-450 dilution refrigerator at a temperature of 15 mK. The input line had an attenuation of 77 dB and the output had more than 60 dB of directional isolation at the cavity and qubit frequencies. The output signal from the cavity was amplified at 3 K by a HEMT amplifier and further amplified at room temperature with a low-noise amplifier. The signal was then mixed down to an intermediate frequency of 10 MHz, digitized and analysed. The present set-up is similar to that described by Novikov et al.[29,30]

**Pulse shaping.** The generation of Gaussian pulses was carried out by programming a Tektronix AWG70002A arbitrary waveform generator (AWG). The maximum sampling rate for the instrument was 25 GS s⁻¹ with 10 bits of resolution for voltage. Numerical simulations were performed in MATLAB using the same pulse shape taking the finite resolution into account (Supplementary Table 1). The output signal of the AWG was amplified at room temperature with a Mini-Circuits ZVA-183-S+ amplifier before being sent into the dilution refrigerator.

**Modelling.** In the numerical simulations, the master equation for the density matrix ($\rho$) was solved in the time domain (Supplementary Tables 1–3). Dissipation of the system was incorporated through the Lindblad–Kossakowski formalism[31,32].

The coupled system of the cavity (dressed resonance at $\omega_{\rm c}$) and the transmon qubit (transition frequency of $\omega_{\rm q}$) is described by the Jaynes–Cummings Hamiltonian,

$$\mathcal{H}_{\rm JC} = \hbar \omega_{\rm q} |e\rangle\langle e| + \hat{\mathcal{N}} \hbar (\omega_{\rm c} + 2\chi |e\rangle\langle e|) \qquad (4)$$

where $\hat{\mathcal{N}}$ is the photon number operator for the cavity and |g⟩, |e⟩ represent the qubit ground and excited states, respectively. The Jaynes–Cummings ladders were truncated to a maximum of 11 levels in the |g⟩ ladder and 10 levels in the |e⟩ ladder.

If the cavity and qubit microwave drives are denoted by $\Omega_{\rm c}(t)$ and $\Omega_{\rm q}(t)$, respectively, the drive Hamiltonian can be written as

$$\mathcal{H}_{\rm int} = \hbar \Omega_{\rm c}(t) \left( a + a^\dagger \right) + \hbar \Omega_{\rm q}(t) (|g\rangle\langle e| + |e\rangle\langle g|) \qquad (5)$$

where $a$ and $a^\dagger$ represent the annihilation and creation operators for the cavity, respectively.

Then the driven system Hamiltonian can be written as, $\mathcal{H} = \mathcal{H}_{\rm JC} + \mathcal{H}_{\rm int}$ and the master equation can be written as,

$$\frac{\partial \rho}{\partial t} = \frac{1}{i\hbar} [\mathcal{H}, \rho] + \sum_j \Gamma_j \left( A_j \rho A_j^\dagger - \frac{1}{2} A_j^\dagger A_j \rho - \frac{1}{2} \rho A_j^\dagger A_j \right) \qquad (6)$$

Here, the index $j$ runs through the various decoherence channels (cavity relaxation, qubit relaxation, qubit dephasing) and $\Gamma_j$ represents the decoherence rate for

each channel. $A_j$ are the Lindblad operators corresponding to each channel and $\rho$ is the density matrix describing the system.

**Reporting fidelity.** Fidelity between any two states (for example realized state and target state) described by density matrices $\rho$ and $\sigma$ is typically defined in quantum information science as

$$F(\rho, \sigma) = \mathrm{Tr}\sqrt{\sqrt{\rho}\sigma\sqrt{\rho}}. \tag{7}$$

The above definition simplifies considerably when one of the states is a pure state (for example $|g1\rangle$ or $|g2\rangle$)[33]. As we were trying to generate pure states, in reporting fidelities of our simulated processes we employed a modified version of the conventional definition to be more intuitive and to concur with the trace distance,

$$\mathcal{F}(\rho, |\psi\rangle) = [F(\rho, |\psi\rangle)]^2 = \langle\psi|\rho|\psi\rangle \tag{8}$$

where $|\psi\rangle$ is the target pure state (for example $|g1\rangle$ or $|g2\rangle$) and $\rho$ is the density matrix for the generated state.

For reporting experimental data, the qubit excited state level was calibrated by performing fast Rabi oscillations and subtracting the background. By checking the amplitude of the Rabi oscillations, the maximum swing of the output signal can be determined. By fitting to a sinusoidal function, the peak value (0.129) was then calibrated to correspond to 100% excited population while the trough value (0.000) was calibrated to correspond to 100% ground population. The signal value at the Rabi peaks can be compared with the expected value from single shots (0.134) performed with the qubit in the excited state (Supplementary Fig. 2). This put a bound of 4% on the state preparation and measurement errors (Supplementary Fig. 3).

**Data availability.** The data that support the findings of this study are available from the corresponding author on request.

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

## Acknowledgements

We thank S. Novikov, J.-H. Yeh, Y. Rosen, F. Strauch and S. Economou for insightful discussions. F.C.W. would like to acknowledge support from the Joint Quantum Institute and the Center for Nanophysics and Advanced Materials.

## Author contributions

S.P.P. designed the experiment, performed the device fabrication, numerical simulations and analysis. S.P.P. and B.S.P. wrote the manuscript. B.S.P. and F.C.W. supervised all aspects of fabrication, experiments and analysis. All authors discussed the results and commented on the manuscript.

## Additional information

**Competing financial interests:** The authors declare no competing financial interests.

**Publisher's note**: 

