## [Peer Review File · Nature Communications]

Reviewers' comments:

Reviewer #1 (Remarks to the Author):

The manuscript by Premaratne et al describes the implementation of a technique to prepare Fock state in a linear cavity exploiting microwave drives and a fixed-frequency qubit.

Previous methods relied on a tunable element. For example, a qubit can be brought in and out of resonance with the cavity [Ref 8 and 9]. One could get similar results using the parametric modulation of either the qubit frequency, the cavity frequency, or the coupling element between them.

To my opinion, the main advantage of the method presented here is that it removes all form of tunability, and therefore avoid sources of decoherence.

Previous implementation of STIRAP were done in highly anharmonic systems. Here, the authors rely solely on the rather small number-dependent dispersive shift between qubit and cavity, and the associated technological challenge is well addressed. The data are clean, match theory very well, and the discussion is clear.

For these reasons I think the manuscript warrant publication in Nature Communication.

I still have a few question/comments:

(1) Paragraph 1 to 3: When offering an alternative to an existing technique, the burden is on the author to describe the pro and cons of the existing techniques, and how the new technique could be better/worse/different. This discussion is missing here. Why should one use STIRAP?

(2) Line 29: In Ref 8 and 9, I believe the qubit was tuned into resonance with the cavity NON-adiabatically compare to the coupling strength (while still adiabatic with respect to the qubit and cavity frequencies)

(3) Line 56: The number of relevant digits seems inconsistent with the rest of the manuscript.

(4) Line 86: A discussion of the origin of δ_1 is missing. I am lacking the physical intuition to understand its effect, and why it takes this or that value.

(5) Measurement: Throughout the paper, the qubit population is measured and, from a fit with theory, the cavity state is inferred. This is pretty convincing because of the high quality of the fit. But why not directly measure the cavity state?

The related discussion in the SI is convoluted, and unfortunately pretty unclear to my opinion. I would suggest to detail that part a little more.

Being in the strong dispersive regime, I believe that qubit spectroscopy could allow the measurement of the Fock state distribution of the cavity. Is there a specific reason that was not done here?

(6) Conclusion: when it comes to further improvement for higher fidelity, the authors give the generic "increase coherence time". This is usually easier said than done. Especially when requiring an order of magnitude increase. The authors suggest to increase σ , but that would compete with the finite coherence time. What about increasing the dispersive shift χ ? Would that allow to shorten the STIRAP protocol, without inducing leakage to other levels?

Reviewer #2 (Remarks to the Author):

The paper employs a circuit QED setup (transmon in a 3D cavity) to create nonclassical states (Fock states in the resonator, superpositions of Fock states, and Bell states). Overall, the paper is well written, at the appropriate scientific level.

However, there are two issues with this work which makes it problematic for publishing in Nat. Commun.

The first is that it is not the first time that Fock states (and almost any combination of them, including superpositions with complex coefficients) have been produced in circuit QED (some of this work is cited in the paper). From this point of view, the results presented do not go beyond the state of the art, although the method is different.

The second issue is that the main claims are not in fact fully substantiated by the results. Indeed, what the authors have measured is the population on the excited state, and showed that this is in agreement with the value expected from simulation. However, this is rather indirect evidence: what they need to show is that the resonator is in a Fock state, superposition of Fock states, and Bell states. Especially for the latter two the situation is more tricky, since it implies demonstrating coherent superpositions. For Bell states for example, measuring just one of the two entangled components is certainly not enough. It seems the authors realized this to some extent and for $n=1$ they did some checkup measurements (in the last section of the Suppl. info). Note also that all the plots presented of P_e show a decrease in P_e as a proof for the creation of the desired state: although this is of course what is expected, one could say that the reduction of population from the excited state can be attributed to anything - including stimulated relaxation. In contrast, previous papers on the same topic have measured the Wigner function, showing that it becomes negative, etc. Here the Mandel Q parameter is calculated not measured.

Other more minor observations: the phase dependence of in Fig. 3 is insufficiently explained - the supplement. info just reproduces the same information and plots. It would require a more detailed analytical understanding. In the absence of this, maybe it is better to remove it (and just leave the plot at zero phase) since it distracts from the main message. Also it would be nice to have a more detailed explanation on how fidelities are calculated using the calibration. Otherwise it is nice that the authors listed all the relevant parameters in the Suppl.

Reviewer #3 (Remarks to the Author):

The authors demonstrate a STIRAP process on a 3D superconducting circuit QED system to generate Fock states and Fock state superpositions as well as Cavity-qubit bell states. This process has so far not been used in circuit QED systems but has been extensively used in cold gases and ion trap experiments. As such it is not a novel technique but might still be interesting in the context of efficiently manipulating the state of a microwave resonator. The authors compare their findings to a master equation simulation of the system and find a very good overlap between theory and experiment. They also investigate the influence of the phase of the initial pulse and the Stokes pulse on the time evolution of the process. The paper is clearly written and the STIRAP process is well explained and discussed in detail.

Nevertheless, having read the paper and the supplementary material I think the paper does not merit a publication in Nature Communications for the following reasons:

- 1.) Why has the experiment been done with an over-coupled cavity? Reducing the output (and maybe even the input) coupling would have increased the total Quality factor of the cavity and thus increased the lifetime. State of the art experiments have shown cavity lifetimes for such a system of $20\mu\text{s}$ and beyond. Such a lifetime would have improved the fidelity of the created states and would have allowed a much more detailed measurement procedure. The authors even mention in the discussion that an increased lifetime would help.

2.)With a higher Q cavity, the authors could have employed a much better analysis of the created cavity states, as has been demonstrated in several papers from the Yale group (e.g. PRL 115, 137002 (2015); PRL 115, 180501 (2015); Nature Communications 6, 8970). Employing similar techniques, the authors could have clearly shown the creation of the different Fock states and measured their fidelities. Even the Bell state of the qubit and the cavity could have been verified. The authors demonstrated a small part of such an analysis in the supplementary material but did not use the methods to their full extend. Especially for the superpositions and the entangled state the authors did not show any coherent properties which in my opinion is necessary to verify the creation of these states. The missing experimental analysis and measurements are the most important point why I do not recommend a publication in Nature Communications.

3.)The authors might want to consider citing PRL 115, 137002 (2015) as the techniques demonstrated there allow for the creation of arbitrary cavity states - including Fock states - in similar times and with similar fidelity.

4.)How much better is the STIRAP process compared to a direct two tone drive, something like a red sideband $|e,0\rangle$ to $|g,1\rangle$ Rabi oscillation (similar to Phys. Rev. B 79, 180511(R))? A direct drive like this would be easier to implement and potentially be faster as it does not have to be adiabatic. Thus it could achieve higher fidelities.

5.)The authors do not mention what the measured excited state population of the qubit and the cavity is. 3D circuit QED setups typically show a few percent excited state population in the qubit and slightly less in the cavity. Is the qubit perfectly in the ground state? Did they use any reset protocols? They also do not discuss the influence of a non-zero qubit excited state population at the beginning of their protocol on the fidelity of the final state.

6.)The authors do not really explain what leads to the overshoot in the excited state population in Figure 3. It matches very well to the theory but there is no explanation what is exactly going on.

In the current state I cannot recommend a publication in Nature Communications. I would recommend that the authors repeat the experiment with a higher Q cavity and use state of the art measurement techniques to verify the created states. At least a simple number state splitting experiment to get experimental data for all theory lines in Figure 2 and Figure 4 should be presented to support the outcome.

Reviewer #1 (Remarks to the Author):

The manuscript by Premaratne et al describes the implementation of a technique to prepare Fock state in a linear cavity exploiting microwave drives and a fixed-frequency qubit.

Previous methods relied on a tunable element. For example, a qubit can be brought in and out of resonance with the cavity [Ref 8 and 9]. One could get similar results using the parametric modulation of either the qubit frequency, the cavity frequency, or the coupling element between them.

To my opinion, the main advantage of the method presented here is that it removes all form of tunability, and therefore avoid sources of decoherence.

Previous implementation of STIRAP were done in highly anharmonic systems. Here, the authors rely solely on the rather small number-dependent dispersive shift between qubit and cavity, and the associated technological challenge is well addressed. The data are clean, match theory very well, and the discussion is clear.

For these reasons I think the manuscript warrant publication in Nature Communication.

I still have a few question/comments:

(1) Paragraph 1 to 3: When offering an alternative to an existing technique, the burden is on the author to describe the pro and cons of the existing techniques, and how the new technique could be better/worse/different. This discussion is missing here. Why should one use STIRAP?

Premaratne et al. respond: Our motivation for doing these experiments was to develop an all microwave technique to create a Fock state. Since the $|e1\rangle$ to $|g0\rangle$ transition is forbidden to first order it seemed natural to try and use STIRAP. Furthermore, a number of research groups in the field are trying to develop techniques to control non-tunable qubit-cavity systems in order to avoid sources of dephasing such as flux noise. As an example, a paper last year (PRL 115, 137002 (2015)) demonstrated the universal control of cavity states by the use of selective number-dependent arbitrary phase (SNAP) gates. The typical con of performing gates using all microwaves is that they are slow (e.g. SNAP gates = 1 μ s, STIRAP gates = 354 ns compared with vacuum Rabi oscillation gates = 35 ns). We have made changes to the introduction so that these pros and cons are indicated.

(2) Line 29: In Ref 8 and 9, I believe the qubit was tuned into resonance with the cavity NON-adiabatically compare to the coupling strength (while still adiabatic with respect to the qubit and cavity frequencies)

Premaratne et al. respond: The word “adiabatically” was removed.

(3) Line 56: The number of relevant digits seems inconsistent with the rest of the manuscript.

Premaratne et al. respond: We changed the reported data values throughout the section to be consistent with each other.

(4) Line 86: A discussion of the origin of δ_1 is missing. I am lacking the physical intuition to understand its effect, and why it takes this or that value.

Premaratne et al. respond: We have added the following sentences to the main text to address this issue. "For a conventional three-level system undergoing STIRAP, maximum process fidelity is attained simply with the Stokes-pump two-photon resonance between the initial and target states and does not depend heavily on the detuning from the intermediate level. However, in our multilevel system, a negative detuning from level e_1 is beneficial to minimize leakage to higher levels and was optimized through simulations." (see paragraph starting at line 100)

(5) Measurement: Throughout the paper, the qubit population is measured and, from a fit with theory, the cavity state is inferred. This is pretty convincing because of the high quality of the fit. But why not directly measure the cavity state?

The related discussion in the SI is convoluted, and unfortunately pretty unclear to my opinion. I would suggest to detail that part a little more.

Being in the strong dispersive regime, I believe that qubit spectroscopy could allow the measurement of the Fock state distribution of the cavity. Is there a specific reason that was not done here?

Premaratne et al. respond: In our current measurement set-up, the noise from the first stage 3K HEMT amplifier in conjunction with the loss in transmission to this amplifier adds a significant number of photon noise.

As you correctly point out we were able to see number splitting in the spectroscopy of the device but that data is taken in the steady state (pumping times $\gg \{T_2, 1/\kappa\}$) and therefore the states generated after the STIRAP protocol would be lost for such spectroscopic measurements. An alternative spectroscopic technique involves a π -pulse from a $|g, n\rangle$ state to $|e, n\rangle$ state before measuring. Because of the lack of photon number anharmonicity in our device, a long weak π -pulse would be required for distinguishing each photon peak accurately and because of the cavity decay rate, we would lose the fidelity of the Fock state in the interim.

The way that the first Fock state was independently verified is by performing resonant (i.e. at a frequency corresponding to $|g_1\rangle \leftrightarrow |e_1\rangle$) and non-resonant (i.e. at a frequency corresponding to $|g_0\rangle \leftrightarrow |e_0\rangle$) Rabi oscillations after the $n=1$ STIRAP protocol. We have moved this discussion from the SI to the main body of the text as well as rewritten it to be clearer. (e.g. see the paragraph at line 128 and figure 3).

(6) Conclusion: when it comes to further improvement for higher fidelity, the authors give the generic "increase coherence time". This is usually easier said than done. Especially when requiring an order of magnitude increase. The authors suggest to increase σ , but that would compete with the finite coherence time. What about increasing the dispersive shift χ_i ? Would that allow to shorten the STIRAP protocol, without inducing leakage to other levels?

Premaratne et al. respond: You are correct. Increasing the dispersive shift χ would allow shorter STIRAP times and in turn higher fidelities. The following statement was added at line 205: “Alternatively, increasing the dispersive shift χ would enable the use of stronger drives and shorter STIRAP times as the increased anharmonicity reduces leakage to other states.”

Reviewer #2 (Remarks to the Author):

The paper employs a circuit QED setup (transmon in a 3D cavity) to create nonclassical states (Fock states in the resonator, superpositions of Fock states, and Bell states). Overall, the paper is well written, at the appropriate scientific level.

However, there are two issues with this work which makes it problematic for publishing in Nat. Commun.

1. The first is that it is not the first time that Fock states (and almost any combination of them, including superpositions with complex coefficients) have been produced in circuit QED (some of this work is cited in the paper). From this point of view, the results presented do not go beyond the state of the art, although the method is different.

Premaratne et al. respond: We have rewritten the introduction on how this demonstration goes beyond state of the art. To summarize, our technique demonstrates very similar state of the art fidelity in a non-tunable system with a smaller anharmonicity and in a shorter gate time.

2. The second issue is that the main claims are not in fact fully substantiated by the results. Indeed, what the authors have measured is the population on the excited state, and showed that this is in agreement with the value expected from simulation. However, this is rather indirect evidence: what they need to show is that the resonator is in a Fock state, superposition of Fock states, and Bell states.

Especially for the latter two the situation is more tricky, since it implies demonstrating coherent superpositions. For Bell states for example, measuring just one of the two entangled components is certainly not enough. It seems the authors realized this to some extent and for $n=1$ they did some checkup measurements (in the last section of the Suppl. info). Note also that all the plots presented of P_e show a decrease in P_e as a proof for the creation of the desired state: although this is of course what is expected, one could say that the reduction of population from the excited state can be attributed to anything - including stimulated relaxation. In contrast, previous papers on the same topic have measured the Wigner function, showing that it becomes negative, etc.

Premaratne et al. respond: Due to the limited photon number anharmonicity and lifetime of the cavity photon states, we were unable to perform Wigner tomography. Instead we demonstrated the fidelity of the generated $n=1$ Fock state by performing resonant (i.e. at a frequency corresponding to $|g1\rangle \leftrightarrow |e1\rangle$) and non-resonant (i.e. at a frequency corresponding to $|g0\rangle \leftrightarrow |e0\rangle$) Rabi oscillations at different drive strengths after the $n=1$ STIRAP protocol. From the extracted Rabi frequencies and amplitude of the oscillations we have determined that the fidelity of the $|n=1\rangle$ Fock state is 85% which is in good agreement with the simulations of 89%.

To address your concerns in the text we have made the following changes:

1. We have made it clearer in the abstract, introduction and elsewhere what is experimentally demonstrated and which values rely on simulations.

2. The discussion of the Rabi oscillations on the $|n=1\rangle$ Fock state has been moved from the SI to the main body of the text and has been expanded for clarification (e.g. see the paragraph at line 128 and figure 3).

3. Based on your comment of the Bell state data, we have removed this discussion and data from the main body of the text.

3. Here the Mandel Q parameter is calculated not measured.

Premaratne et al. respond: You are correct. As the Mandel Q parameter only reinterprets existing information, it has been removed.

4. Other more minor observations: the phase dependence of in Fig. 3 is insufficiently explained - the supplement. info just reproduces the same information and plots. It would require a more detailed analytical understanding. In the absence of this, maybe it is better to remove it (and just leave the plot at zero phase) since it distracts from the main message.

Premaratne et al. respond: We have followed your advice and moved all of the plots except the zero phase plot to the supplemental section. We have replaced the figure on the right with the evolution of an incoherent state.

5. Also it would be nice to have a more detailed explanation on how fidelities are calculated using the calibration.

Premaratne et al. respond: We have included a new subsection in the Methods section at line 264 to address this comment.

6. Otherwise it is nice that the authors listed all the relevant parameters in the Suppl.

Reviewer #3 (Remarks to the Author):

The authors demonstrate a STIRAP process on a 3D superconducting circuit QED system to generate Fock states and Fock state superpositions as well as Cavity-qubit bell states. This process has so far not been used in circuit QED systems but has been extensively used in cold gases and ion trap experiments. As such it is not a novel technique but might still be interesting in the context of efficiently manipulating the state of a microwave resonator. The authors compare their findings to a master equation simulation of the system and find a very good overlap between theory and experiment. They also investigate the influence of the phase of the initial pulse and the Stokes pulse on the time evolution of the process. The paper is clearly written and the STIRAP process is well explained and discussed in detail.

Nevertheless, having read the paper and the supplementary material I think the paper does not merit a publication in Nature Communications for the following reasons:

1.) Why has the experiment been done with an over-coupled cavity? Reducing the output (and maybe even the input) coupling would have increased the total Quality factor of the cavity and thus increased the lifetime. State of the art experiments have shown cavity lifetimes for such a system of 20 μ s and beyond. Such a lifetime would have improved the fidelity of the created states and would have allowed a much more detailed measurement procedure. The authors even mention in the discussion that an increased lifetime would help.

Premaratne et al. respond: Our single cavity –qubit system was over-coupled to measure the state of the system. At line 86 we have added the text “for purposes of measurement”.

We agree with you, the next step in improving the Fock state fidelity is to manufacture a two cavity system where an under-coupled cavity would be used to store the photon.

2.) With a higher Q cavity, the authors could have employed a much better analysis of the created cavity states, as has been demonstrated in several papers from the Yale group (e.g. PRL 115, 137002 (2015); PRL 115, 180501 (2015); Nature Communications 6, 8970). Employing similar techniques, the authors could have clearly shown the creation of the different Fock states and measured their fidelities. Even the Bell state of the qubit and the cavity could have been verified. The authors demonstrated a small part of such an analysis in the supplementary material but did not use the methods to their full extent.

Especially for the superpositions and the entangled state the authors did not show any coherent properties which in my opinion is necessary to verify the creation of these states. The missing experimental analysis and measurements are the most important point why I do not recommend a publication in Nature Communications.

Premaratne et al. respond: Due to the limited photon number anharmonicity and lifetime of the cavity photon states, we were unable to perform Wigner tomography. Instead we demonstrated the fidelity of the generated $n=1$ Fock state by performing resonant (i.e. at a frequency corresponding to $|g1\rangle \leftrightarrow |e1\rangle$) and non-resonant (i.e. at a frequency corresponding to $|g0\rangle \leftrightarrow |e0\rangle$) Rabi oscillations at different

drive strengths after the $n=1$ STIRAP protocol. From the extracted Rabi frequencies and amplitude of the oscillations we have determined that the fidelity of the $|n=1\rangle$ Fock state is 85% which is in good agreement with the simulations of 89%.

To address your concerns in the text we have made the following changes:

1. We have made it clearer in the abstract, introduction and elsewhere what is experimentally demonstrated and which values rely on simulations.
2. The discussion of the Rabi oscillations on the $|n=1\rangle$ Fock state has been moved from the SI to the main body of the text and has been expanded for clarification (e.g. see the paragraph at line 127 and figure 3).
3. We have removed the Bell state discussion and data from the main body of the text.

3.)The authors might want to consider citing PRL 115, 137002 (2015) as the techniques demonstrated there allow for the creation of arbitrary cavity states - including Fock states - in similar times and with similar fidelity.

Premaratne et al. respond: Thank you for pointing this out. We have included the reference in the text.

4.)How much better is the STIRAP process compared to a direct two tone drive, something like a red sideband $|e,0\rangle$ to $|g,1\rangle$ Rabi oscillation (similar to Phys. Rev. B 79, 180511(R))? A direct drive like this would be easier to implement and potentially be faster as it does not have to be adiabatic. Thus it could achieve higher fidelities.

Premaratne et al. respond: You are correct in asserting the quicker process of sideband drive. To address your concerns we have explained the advantages of STIRAP in the introduction, the elaborated on the sideband driving in the discussion, and included the references PRB 79, 180511 and PRL 99, 050501 as well.

5.)The authors do not mention what the measured excited state population of the qubit and the cavity is. 3D circuit QED setups typically show a few percent excited state population in the qubit and slightly less in the cavity. Is the qubit perfectly in the ground state? Did they use any reset protocols? They also do not discuss the influence of a non-zero qubit excited state population at the beginning of their protocol on the fidelity of the final state.

Premaratne et al. respond: Apart from passive cooling, no active reset protocols were employed in this experiment.

An upper bound for the cavity population of $< 2\%$ was extracted from number splitting measurements. We have placed an upper bound of $< 4\%$ from qubit spectroscopy at the e_0 to f_0 (i.e. second excited state) spectroscopic transition where we didn't observe a clear peak.

We studied the effect of residual populations in the qubit and cavity through simulations.

- For $n=1$ Fock state generation, a residual qubit excited population would be transferred to the ground state after the initialization pulse. Since the $|g_0\rangle$ population is unaffected by STIRAP, it will remain there. Hence, the residual population would lower the fidelity by an equal amount.
- For $n=1$ Fock state generation, a residual cavity excited population would be partially transferred to $|e_1\rangle$. This will also result in a loss of fidelity by an equal amount to the initial residue.

We have added a paragraph to the Supplementary material addressing this.

6.) The authors do not really explain what leads to the overshoot in the excited state population in Figure 3. It matches very well to the theory but there is no explanation what is exactly going on.

Premaratne et al. respond: We added the paragraph “To further understand the effect of the initial state...” at the end of the section **Superposition of Fock states** to address your concern.

In the current state I cannot recommend a publication in Nature Communications. I would recommend that the authors repeat the experiment with a higher Q cavity and use state of the art measurement techniques to verify the created states. At least a simple number state splitting experiment to get experimental data for all theory lines in Figure 2 and Figure 4 should be presented to support the outcome.

Premaratne et al. respond: We hope the corrections made and the additional material added is enough to convince you of the highly coherent nature of the states generated.

Reviewers' comments:

Reviewer #1 (Remarks to the Author):

I am satisfied by the author's response and by the revised manuscript.
I recommend publication of this manuscript in Nature Communication.

Reviewer #2 (Remarks to the Author):

The resubmitted version includes more material and some clarifications. The authors clearly has made the effort to reorganize some ideas - and I can only imagine how high is the work load for an experimental paper with only three authors.

These being said, I am unfortunately not convinced that the paper is suitable for an interdisciplinary high-profile journal. The main trouble arises from the novelty of the approach, as well as the verification of the claims. The creation of states with highly non-classical properties has been demonstrated before in similar systems, and the verification of these properties has been done in a better way. Some of these works are already cited, but there are more groups (e.g. Wallraff at ETH or Huard at ENS) who have either explored the upper states of the Jaynes-Cummings model or who used the ac Shift of single photons in the cavity. Fair enough, the authors use a new method (STIRAP) to create these states, but that's pretty much it. Again, if I compare the main claims of the paper with what has been achieved so far in similar systems, I can see reasonable indirect evidence for Fock states (the fact that the qubit transitions become ac-Stark shifted) but the evidence for coherence in the superpositions of Fock states is rather slim. Unfortunately in the sample studied it is not possible to find evidence for the nonclassicality of either of these states, as has been done by people who measured the Wigner function.

Reviewer #3 (Remarks to the Author):

The authors have addressed most of the issues raised in the initial report. Important information has been move from the supplementary material to the main text. The data in Fig. 3 now shows that they successfully prepare a single photon state with a fidelity close to 90%. Also Fig. 4 now demonstrates that the preparation of the Fock state superposition is coherent.

The two remaining issues I have with the papers are:

- The authors do not use state of the art methods to analyze the created states. I understand that this is due to a limitation of this particular experiment. Still I wonder why the authors did not consider using a state of the art multi cavity system (or even employing multiple modes of the single 3D cavity) right from the start to investigate a method to prepare Fock states and superpositions of Fock states. This would have allowed them to get a combination of good coherence times and high enough readout fidelity.
- The comment regarding the sideband driving has only been partially addressed. I still wonder whether the achievable fidelities with a sideband drive are higher than the STIRAP fidelities, due to the reduced time. As both cases are equally simply in the implementation, I think the higher fidelity process would be the one to go for.

Reviewers' comments:

Reviewer #1 (Remarks to the Author):

I am satisfied by the author's response and by the revised manuscript.
I recommend publication of this manuscript in Nature Communication.

Reviewer #2 (Remarks to the Author):

The resubmitted version includes more material and some clarifications. The authors clearly has made the effort to reorganize some ideas - and I can only imagine how high is the work load for an experimental paper with only three authors.

These being said, I am unfortunately not convinced that the paper is suitable for an interdisciplinary high-profile journal. The main trouble arises from the novelty of the approach, as well as the verification of the claims. The creation of states with highly non-classical properties has been demonstrated before in similar systems, and the verification of these properties has been done in a better way. Some of these works are already cited, but there are more groups (e.g. Wallraff at ETH or Huard at ENS) who have either explored the upper states of the Jaynes-Cummings model or who used the ac Shift of single photons in the cavity. Fair enough, the authors use a new method (STIRAP) to create these states, but that's pretty much it. Again, if I compare the main claims of the paper with what has been achieved so far in similar systems, I can see reasonable indirect evidence for Fock states (the fact that the qubit transitions become ac-Stark shifted) but the evidence for coherence in the superpositions of Fock states is rather slim. Unfortunately in the sample studied it is not possible to find evidence for the nonclassicality of either of these states, as has been done by people who measured the Wigner function.

Premaratne et al. respond: The focus of our work was to implement a new all microwave technique to create Fock states. From that standpoint we believe this work is unique. The work led by Wallraff in creating a Bell state between a qubit and itinerant photon (PRL 109, 240501 (2012)) used a tunable qubit to create their non-classical states similar to the work in reference 8 and 9 of our manuscript. The work led by Huard (PRL 109, 183901 (2012)) in creating entangled itinerant photons used Josephson mixers, which in itself is very novel. From our perspective the implementation scheme we demonstrate in this manuscript is simpler and has state of the art fidelity.

At the onset of this experiment, independent techniques (e.g. Heeres, et al. PRL **115**, 137002 (2015)) to analyze the stored generated Fock states were not as prevalent and have only come to the forefront recently. We directly address this point in the manuscript with the following comment in the manuscript on line 202 of the discussion "...and allow the application of more sophisticated analysis techniques on the generated states [10]...

Reviewer #3 (Remarks to the Author):

The authors have addressed most of the issues raised in the initial report. Important information has been moved from the supplementary material to the main text. The data in Fig. 3 now shows that they successfully prepare a single photon state with a fidelity close to 90%. Also Fig. 4 now demonstrates that the preparation of the Fock state superposition is coherent.

The two remaining issues I have with the papers are:

- The authors do not use state of the art methods to analyze the created states. I understand that this is due to a limitation of this particular experiment. Still I wonder why the authors did not consider using a state of the art multi cavity system (or even employing multiple modes of the single 3D cavity) right from the start to investigate a method to prepare Fock states and superpositions of Fock states. This would have allowed them to get a combination of good coherence times and high enough readout fidelity.

Premaratne et al. respond: We understand the reviewer's argument regarding state-of-the-art measurement techniques. At the onset of this experiment, independent techniques (e.g. Heeres, et al. PRL **115**, 137002 (2015)) to analyze the generated states were not prevalent. These techniques came to the forefront within the past two years. We directly address this point in the manuscript with the following comment in the manuscript on line 202 of the discussion "...and allow the application of more sophisticated analysis techniques on the generated states [10]..."

Reviewer 3 Question: The comment regarding the sideband driving has only been partially addressed. I still wonder whether the achievable fidelities with a sideband drive are higher than the STIRAP fidelities, due to the reduced time. As both cases are equally simple in the implementation, I think the higher fidelity process would be the one to go for.

Premaratne et al. respond: As you suggest, decreasing the operation time is a potential way of improving upon the fidelity, assuming the technique is accurate. To determine if a sideband drive technique is accurate and has a higher fidelity, we performed extensive density matrix simulations of the $|e0\rangle \rightarrow |g1\rangle$ sideband transitions using a master equation approach and have worked to optimize the target fidelity. We find that the fidelity using sideband transitions ($\mathcal{F} \approx 60\%$) is significantly lower compared with STIRAP. Note that to increase the fidelity to this value, we had to increase the time of operation (see Fig. A for the populations of the five lowest levels as a function of time). Utilizing this technique also results in rapid oscillations/beating in the state populations. To obtain fidelities similar to what we have observed in this manuscript, we would have to increase the dispersive shift to $\chi/2\pi = -70$ MHz (see Fig. B). Understanding these differences between the two techniques is outside the scope of this work. This has been addressed starting at line 144 as "In comparison, when simulating a 2-photon red sideband..."

Figure A: Sideband oscillations between e0 and g1 with $\chi/2\pi = -2 \text{ MHz}$

Figure B: Sideband oscillations between e0 and g1 with $\chi/2\pi = -70 \text{ MHz}$

Finally, we have recently learned about a few proposals and demonstrations of shortcuts to adiabaticity, which could be another way of speeding up the operation time. We have expanded upon this in the discussion with the following paragraph beginning at line 207 in the discussion for future work: “Implementing adiabatic shortcuts [24–26] is another approach which could potentially speed up the transfer process in our system”

REVIEWERS' COMMENTS:

Reviewer #3 (Remarks to the Author):

The resubmitted version has added an analyses of creating a one photon Fock state with a sideband drive to the reply, which I think should at least be added to the supplementary methods of the paper. The parameter range explored for this simulation are not really given, still the authors state that they have tried to optimize the fidelity. This analyses addresses my question raised in the last report, whether a sideband drive would be better suited. From the simulations it is clear that, in a system like theirs, the RAP shows a much higher fidelity. A small question remains whether things would be reversed if the coherence time of the cavity would be higher – although I doubt it.

The last remaining issue I have with the paper is that the authors did not really address why they did not use state of the art methods to analyze the created states. Especially I do not agree with their statement in the reply that these analysis techniques where not prevalent. The mehtods were used in several papers by the Yale group starting in 2013 (Nature 495, 205–209, (2013)) – so about three years ago. The PRL 115 is just another one in a series of papers using these techniques. So I would call them rather well established especially as they were already used by other groups e.g. Huard (Science Vol. 348, Issue 6236, pp. 776-779, (2015)).

Nevertheless, I think that the authors have shown enough evidence of creating the intended states especially including the supplementary material and the comparison of the Rabi oscillations with various drive strength to the theory.

Even though I do not have any objections in general publishing the paper I am not convinced that nature communications is the right journal to do so. The issues are, that the approach is only somewhat novel (RAP is a well-established method in e.g. AMO – here it is just a little bit more complicated to implement due to the higher lying states) and the analysis is not really state of the art.

REVIEWERS' COMMENTS:

Reviewer #3 (Remarks to the Author):

The resubmitted version has added an analyses of creating a one photon Fock state with a sideband drive to the reply, which I think should at least be added to the supplementary methods of the paper. The parameter range explored for this simulation are not really given, still the authors state that they have tried to optimize the fidelity. This analyses addresses my question raised in the last report, whether a sideband drive would be better suited. From the simulations it is clear that, in a system like theirs, the RAP shows a much higher fidelity. A small question remains whether things would be reversed if the coherence time of the cavity would be higher – although I doubt it.

We have included the analysis of the red sideband transitions in the Supplementary information section.

The last remaining issue I have with the paper is that the authors did not really address why they did not use state of the art methods to analyze the created states. Especially I do not agree with their statement in the reply that these analysis techniques where not prevalent. The mehtods were used in several papers by the Yale group starting in 2013 (Nature 495, 205–209, (2013)) – so about three years ago. The PRL 115 is just another one in a series of papers using these techniques. So I would call them rather well established especially as they were already used by other groups e.g. Huard (Science Vol. 348, Issue 6236, pp. 776-779, (2015)).

We have included in the discussion section a sentence describing the limitations of our system in using the state-of-the-art methods, which were not in the forefront at the time the work was initiated. Moving forward, we thank the reviewer for being objective on the need for better measurement techniques.

Nevertheless, I think that the authors have shown enough evidence of creating the intended states especially including the supplementary material and the comparison of the Rabi oscillations with various drive strength to the theory.

Even though I do not have any objections in general publishing the paper I am not convinced that nature communications is the right journal to do so. The issues are, that the approach is only somewhat novel (RAP is a well-established method in e.g. AMO – here it is just a little bit more complicated to implement due to the higher lying states) and the analysis is not really state of the art.